# Lignans Extract from Knotwood of Norway Spruce—A Possible New Weapon against GTDs

**DOI:** 10.3390/jof8040357

**Published:** 2022-03-30

**Authors:** Milan Špetík, Josef Balík, Pavel Híc, Eliška Hakalová, Kateřina Štůsková, Lucie Frejlichová, Jan Tříska, Aleš Eichmeier

**Affiliations:** 1Mendeleum—Institute of Genetics, Faculty of Horticulture, Mendel University in Brno, Valtická 334, 691 44 Lednice na Moravě, Czech Republic; eliska.hakalova@mendelu.cz (E.H.); xstusko1@mendelu.cz (K.Š.); xfrejlic@mendelu.cz (L.F.); ales.eichmeier@mendelu.cz (A.E.); 2Department of Post-Harvest Technology of Horticultural Products, Faculty of Horticulture, Mendel University in Brno, Valtická 334, 691 44 Lednice na Moravě, Czech Republic; josef.balik@mendelu.cz (J.B.); pavel.hic@mendelu.cz (P.H.); 3Global Change Research Institute CAS, Bělidla 986/4a, 603 00 Brno, Czech Republic; triska.j@czechglobe.cz

**Keywords:** bioprotection, GTD, grapevine, HMR, 7-hydroxymatairesinol, wood extract, Norway spruce

## Abstract

Grapevine trunk diseases (GTDs) pose a major threat to the wine industry worldwide. Currently, efficient biological methods or chemical compounds are not available for the treatment of infected grapevines. In the present study, we used an extract from the knotwood of spruce trees as a biological control against GTDs. Our in vitro trial was focused on the antifungal effects of the extract against the most common GTD pathogens—*Cadophora luteo-olivacea*, *Dactylonectria torresensis*, *Diaporthe ampelina*, *Diaporthe bohemiae*, *Diplodia seriata*, *Eutypa lata*, and *Phaeoacremonium minimum*. Our in vitro trial revealed a high antifungal effect of the extract against all tested fungi. The inhibition rates varied among the different species from 30% to 100% using 1 mg·mL^−1^ extract. Subsequently, the efficiency of the extract was supported by an in planta experiment. Commercial grafts of *Vitis vinifera* were treated with the extract and planted. The total genomic DNA of grapevines was extracted 10 days and 180 days after the treatment. The fungal microbial diversities of the treated/untreated plants were compared using high-throughput amplicon sequencing (HTAS). Treated plants showed 76.9% lower relative abundance of the genus *Diaporthe* and 70% lower relative abundance of the genus *Phaeoacremonium* 10 days after treatment. A similar scenario was observed for the genus *Cadophora* 180 days after treatment, where treated plants showed 76% lower relative abundance of this genus compared with untreated grapevines.

## 1. Introduction

The grapevine (*Vitis vinifera* L.) is an important agricultural crop with a significant economic impact worldwide. In 2018, world vineyards covered an area of approximately 7.4 million hectares, producing 78 million tons of grapes. In the same year, world trade in wine (not including table grapes and raisins) accounted for 108 million hectolitres in volume and EUR 31 billion in value [1]. Grapevine trunk diseases (GTDs) pose a major threat to viticulture, causing serious economic losses to the sector [2]. Annual costs for replacing vineyards damaged by GTDs are estimated to be approximately USD 1.5 million [3]. GTDs are a group of grapevine diseases caused by several fungal pathogens that colonize woody tissues. Symptoms include wood necrosis, wood discolouration, vascular infections, decay, and foliar symptoms [4,5]. Affected vines show a general and progressive decline that can initially cause the loss of productivity and often the death of the vine. Currently, approximately 130 GTD pathogens are known. Some of the most destructive GTDs are ESCA proper, Eutypa dieback, Botryosphaeria dieback, black foot disease, and Phomopsis cane and leaf spot [6,7,8,9].

Since sodium arsenite was banned in 2001 due its negative impact on the environment, no efficient treatment is available to control GTDs [5,10]. It remains challenging for scientists all over the world to find effective protection/treatment against this complex range of diseases. The solution for the current situation could be a lignan extract from the knotwood of *Picea abies*, which has been give the working name “green extract (GE)”. Recently, several authors described the antimicrobial ability of wood extracts against various fungal species belonging to the genera *Aspergillus*, *Colletotrichum*, *Fusarium*, *Gloeophyllum*, *Mycosphaerella*, *Pycnoporus*, *Penicillium*, *Phytophthora*, *Schizophyllum*, and *Trametes* [11,12,13,14]. Knotwood extracts from *P. abies* contain the 7-hydroxymatairesinol (HMR) lignan as the predominant compound [15,16]. The HMR lignan is known to have antifungal abilities itself [16,17,18]. In the present work, we studied the antifungal abilities of GE against GTDs both in vitro and in planta.

We tested the following hypotheses: (i) GE has antifungal abilities against GTD fungal pathogens in vitro; (ii) HMR lignan, which is the dominant compound of GE, has antifungal properties against GTD fungi itself; (iii) the grapevine propagation material is already infected by GTD pathogens in nurseries; and (iv) if GE provides antifungal activity against GTD fungi in vitro, then the same scenario will be observed against GTD fungi inside young grapevine plants.

## 2. Materials and Methods

### 2.1. Fungal Strains

For the in vitro testing, we used seven fungal strains—*Cadophora (C.) luteo-olivacea* (J.F.H. Beyma) of T.C. Harr. and McNew (MEND-F-0454); *Dactylonectria (Da.) torresensis* (A. Cabral, Rego, and Crous) of L. Lombard and Crous (MEND-F-0016); *Diaporthe (D.) ampelina* (Berk. and M.A. Curtis) of R.R. Gomes, Glienke, and Crous (MEND-F-0157); *Diaporthe (D.) bohemiae* of Guarnaccia, Eichmeier, and Crous (CBS 143347); *Diplodia (Dip.) seriata* of De Not. (MEND-F-0014); *Eutypa (E.) lata* (Pers.) of Tul. and C. Tul. (MEND-F-0015); *Phaeoacremonium (P.) minimum* (Tul. and C. Tul.) of Gramaje, L. Mostert, and Crous (MEND-F-0013)—from (MEND-F), Fungal Culture Collection of Mendeleum, Mendel University in Brno, and (CBS) Westerdijk Fungal Biodiversity Institute, The Netherlands.

### 2.2. Preparation of the GE

Green extract was obtained using the method described by [19] from the knotwood of *P. abies*. Briefly, the knot chips were boiled in water three times for 60 min. The decanted and combined extracts were evaporated and then lyophilized. The obtained dried extract was dissolved in alcohol (96% *w*/*w*) and concentrated by evaporation after removal of the insoluble fractions by filtration. The final extract contained HMR—110.5 mg·mL^−1^ and alpha-conidendrin (coni)—8.4 mg·mL^−1^, solute in 96% (*w*/*w*) ethanol.

### 2.3. In Vitro Antifungal Activity of GE

The first experiment was performed in vitro by testing GE directly against selected pathogenic GTD fungi. The antifungal activity assays were performed using 40 mm-diameter glass Petri dishes, containing potato dextrose agar (PDA, HiMedia, Mumbai, India) supplemented with GE. The GE was added into the media before the autoclaving step. HMR lignan is the main antimicrobial compound of GE [17]. Thus, three different concentrations of GE in the medium were tested based on the HMR content, containing 0.1, 0.5, and 1 mg·mL^−1^ HMR in the final plates. The supplemented plates were inoculated with 5 mm plugs from 10-day-old cultures of representative pathogenic fungi. Subsequently, the plates were sealed with Parafilm© (Bemis Europe, Braine L’Alleud, Belgium) and incubated at 25 °C in the dark. The colony diameters of for each plate were measured after five days of incubation for all tested fungi except *C. luteo-olivacea* and *P. minimum*, for which the measurements were performed after 10 days because of their slow growth. To eliminate the solvent effect, PDA plates supplemented with ethanol in the same amounts as in the GE plates were used for controls. For each concentration, five plates in three repetitions were used. The entire experiment was performed twice. An example for *E. lata* is shown in Appendix A. The inhibition indices were calculated by the following formula: (I) = C − T/C × 100; where C is the growth (diam) of the pathogen in the control plates, T is the growth (diam) of the pathogen in plates supplemented with GE, and I is the percent inhibition of mycelial growth.

### 2.4. In Vitro Antifungal Activity of Pure HMR vs. GE

In addition to the HMR lignan, GE also contains other compounds that could have antifungal effects, such as other lignans, oligolignans, flavonoids, or stilbenes [16,20]. To evaluate whether HMR is the main antifungal compound of GE, the same experiment, as that using GE, was performed using pure HMR lignan (HMRlignan^TM^) instead of GE. HMRlignan^TM^ was provided by Linnea SA (Locarno, Switzerland). For each concentration, five plates in three repetitions were used. The entire experiment was performed twice. The inhibition indices were calculated in the same way as for the GE trial. Subsequently, the results for GE and HMRlignan^TM^ were compared.

### 2.5. Plant Material and Treatments

Based on promising in vitro studies, we performed an experiment in planta. The experiment was carried out during spring–autumn 2019 to examine the effect of GE on grapevine mycobiota. One hundred grapevines cv. Sauvignon Blanc grafted on Kober 5BB rootstock were purchased from a Czech commercial nursery. Based on a previous experiment, we assumed that the plants purchased from the nursery were already naturally infected by GTD pathogens [21]. The plants were randomly divided into two bundles—50 plants/bundle. The first bundle was treated with GE, diluted using distilled water to contain 1 mg·mL^−1^ HMR lignan. The use of the concentration of 1 mg·mL^−1^ HMR lignan in the treatment was selected based on the in vitro results, where this concentration significantly inhibited the growth of the tested fungi. The second bundle was treated with distilled water supplemented with ethanol 96% (*w*/*w*) in the same amount as that used in the GE treatment and thus served as controls. The treatments were applied in plastic pots containing treatment agents above the root mass level for a period of 24 h. Both treated and untreated (control) plants were planted directly into the soil and cultivated in the field conditions.

### 2.6. Extraction of the DNA and DNA Pooling

The total genomic DNA extractions from the plant material s were performed at Mendel University in Brno at 10 days after treatment (evaluating the immediate effects of GE) and at 180 days after treatment (evaluating the effects of GE after one vegetative growth period). From each treatment and each period, 18 plants were randomly selected for DNA extraction: 2 treatments (untreated/treated) × 2 periods (10 days/180 days) × 18 plants = 72 plants in total. The plants were washed with tap water to remove residual soil, washed with sterile distilled water, and dried. Two parts from each plant were used for each DNA sample: the grafting area and the basal end of the rootstock. Each plant part was debarked, and its wood was scraped with a sterile scalpel under sterile conditions. One hundred milligrams of wood was collected and ground in a mortar cooled to −80 °C, and 50 mg of this homogenate was used for DNA extraction using a Macherey Nucleospin Tissue kit (Macherey-Nagel, Düren, Germany). The total DNA yield was measured using a fluorimeter and diluted to 10 ng·μL^−1^. Subsequently, the DNA samples were pooled. Three samples of DNA from the same treatment and same period were pooled into one DNA sample. The 72 DNA samples were pooled into 24 final samples that were further used for amplicon library preparation.

### 2.7. Library Preparation and Sequencing

The complete transcribed spacer region [22] was amplified using the barcoded primers ITS1 and ITS4 [23]. PCR was carried out in 50 μL reaction volumes using 25 μL of Q5^®^ High-Fidelity 2× Master Mix (NEB, Ipswich, UK), 2.5 µL of each primer (10 µM), 2 μL of template DNA, and 18 µL of nuclease-free water. An initial denaturation step of 2 min at 95 °C was followed by amplification for 35 cycles under the following conditions: 30 s at 95 °C, 30 s at 55 °C, and 60 s at 72 °C. A final 5 min extension at 72 °C completed the protocol. The final PCR products were separated on a 1.3% agarose gel (Serva, Heidelberg, Germany) and purified with a NucleoSpin Tissue kit (Macherey-Nagel, Düren, Germany) according to the manufacturer’s instructions. The cDNA amplicon library was prepared according to the Illumina Nextera XT DNA Library Prep protocol (Illumina, San Diego, CA, USA). The final library was subjected to a quality check using a Fast qPCR Library Quantification Kit (MCLAB, San Francisco, CA, USA) and sequenced using MiniSeq (Illumina) (2 × 150 base reads) with a MiniSeq Mid Output Kit (300 cycles) (Illumina). Negative controls were included during the extraction, amplification, and sequencing to evaluate potential contamination throughout the entire process.

### 2.8. Bioinformatic and Data Evaluation

Sequence quality was visualized using FastQC-0.10.1 [24]. Further data processing was completed using SEED v2.1.2 [25]. Raw paired-end reads were joined using fastq-join [26]. The amplicon library was trimmed and clustered, and the sequences containing ambiguities were discarded as well as those with average read quality values lower than Q30. Subsequently, specific primers and sequences smaller than 100 bp were removed. The ITS2 regions were extracted using ITSx implemented in SEED2 [27]. Then, chimeric sequences were discarded using UPARSE implemented in USEARCH v8.1.1861 [28], and the data were clustered into operational taxonomic units (OTUs) using the same tool at 97% similarity. The most abundant OTUs were classified using UNITE v 8.2 [29]. Sequences identified as nonfungal were discarded. The HTAS data were deposited as SRA archives in NCBI GenBank under BioProject Acc. No. PRJNA746244.

### 2.9. Statistical Analyses of In Vitro Experiment

The assumption of data normality was checked, and it met the requirements for parametric analysis. Comparisons of in vitro fungal colony growth rates were performed using one-way ANOVA. Post hoc analyses were performed where required using Tukey’s multiple comparisons test. The significance standard for all tests was set at *p* ≤ 0.05. Statistical analyses were performed using Statistica12© StatSoft software (Tibco Software, Palo Alto, CA, USA).

### 2.10. Analyses of HTS Data

OTUs with fewer than five reads in total were removed from the library [30]. Subsequently, the filtered OTU table (Appendix A) was normalized using rarefaction into 11,129 reads per sample to remove sample heterogeneity. The rarefied OTU table was used to calculate alpha diversity indices, including Shannon diversity and Chao1 richness, using the Phyloseq package as a tool in MicrobiomeAnalyst [31]. Alpha diversities were compared using one-way ANOVA and Tukey’s multiple comparison test. The fungal community compositions were evaluated using principal coordinate analysis (PCoA) plots of Bray–Curtis distances using MicrobiomeAnalyst. Heatmaps were employed to visualize the prevalence of fungal genera within both treatments and sampling periods using the same tool. Differentially abundant OTUs were identified using linear discriminant analysis effect size (LEfSe) at the genus level in MicrobiomeAnalyst. The Wilcoxon *p* value was set at 0.05, and the linear discriminant analysis (LDA) threshold score was set at 2.0. Rarefaction curves and Good’s coverage values were calculated using MicrobiomeAnalyst. The fungal OTUs shared between treatments were visualized using a Venn diagram analysis (http://bioinformatics.psb.ugent.be accessed on 10 January 2022).

## 3. Results

### 3.1. GE Exhibits Strong Antifungal Effects against GTD Fungi In Vitro

In vitro trials of GE have revealed antifungal effects against all tested fungi (Figure 1). GE plates containing 0.1 mg·mL^−1^ HMR in medium had significant inhibitory effects against *C. luteo-olivacea Da. torresensis*, *D. ampelina*, *D. bohemiae*, and *E. lata* compared with the control. The inhibition of growth varied per species from 25.8 to 47.5%. No inhibitory effect was observed against *Dip. seriata* and *P. minimum* using 0.1 mg·mL^−1^ of HMR. The second tested concentration of GE containing 0.5 mg·mL^−1^ HMR in medium inhibited the growth of all tested fungi, including *Dip. seriata* and *P. minimum.* The inhibition per species ranged from 26.6 to 65.3%. The third tested concentration of GE containing 1 mg·mL^−1^ HMR in the medium was the most effective, inhibiting the growth of all the tested fungi. The inhibition per different species varied from 30 to 100%. The lowest growth inhibition (30%) was observed against *Dip. seriata.* Total growth inhibition (100%) was observed against *C. luteo-olivacea*, *Da. Torresensis*, and *P. minimum.* The detailed results of the inhibition observed per fungus in percent are listed in (Appendix A).

### 3.2. In Vitro Trials Reveal Stronger Antifungal Capacity of GE Compared to Pure HMR^TM^

In the trial examining the inhibition effects of GE plates containing 1 mg·mL^−1^ HMR against pure HMR^TM^ plates containing 1 mg·mL^−1^ HMR^TM^, the inhibition effects of HMR^TM^ on fungal growth were lower than those of the GE plates, as follows: *C. luteo-olivacea*—53%; *Da. Torresensis*—28.9%; *Dip. seriata*—25%; *E. lata*—7%; *P. minimum*—73.4%. In the case of *D. ampelina*, the inhibition was 5% stronger on HMR^TM^ plates than on GE plates (67.5% on HMR^TM^ × 62.5% on GE). The results for *D. bohemiae* were similar (58.1% on HMR^TM^ × 58.4% on GE). Complete results are shown in (Figure 2) and as a percentage in (Appendix A).

### 3.3. Sequencing Depth and Community Diversity

A total of 841,546 high-quality ITS2 sequences from 24 samples were retained after pairing ends, quality filtering, ITS2 extraction, and singleton/chimer removal. The minimum, maximum, and average reads per sample were 2835, 67,648, and 35,064, respectively (Appendix A). One sample (GE10b1) was excluded from the analyses due to a low read count (2835 reads). A total of 273 unique OTUs were identified. No contamination was detected in the negative control used in the DNA extraction and amplification step. Good’s coverage values ranged from 99.9 to 100% (Appendix A), and together with rarefaction curves (Appendix A), showed that our sampling covered all the diversity at an adequate sequencing depth. The Chao1 richness estimator ranged from 24.3 to 45, and the Shannon diversity estimator ranged from 1.52 to 2.56 (Appendix A).

### 3.4. Taxonomic Distribution of Fungi Identified by HTAS

In the dataset combining both sampling periods, ten classes belonging to three phyla were identified. The predominant phyla were Ascomycota (96.7%) and Basidiomycota (3.3%). Ascomycota had the highest number of reads in each sample, with a median of 99.4% of the total fungal population within all samples. The relative abundances of different phyla, families, and genera are shown in Appendix A.

The alpha diversity metrics of the fungal communities did not differ significantly (*p* > 0.05; Appendix A). However, the principal coordinates analysis (PCoA) of Bray–Curtis data demonstrated that the sampling period was the primary source of beta diversity (*p* < 0.002; Figure 3). Therefore, we decided to analyse both sampling periods separately.

The most abundant genera (Figure 4) **10 days after planting were found in the following**
**untreated plants**: *Diaporthe* (19.34%), *Notophoma* (17.34%), *Cadophora* (10.68%), *Alternaria* (7.13%), *Ectophoma* (5.19%), *Sporothrix* (5.05%), *Phialocephala* (4.86%), *Trichoderma* (4.24%), *Fusarium* (3.65%), *Phaeoacremonium* (3.56%), and *Stenocarpella* (2.85%). The most abundant genera **ten days after planting were found in the following treated plants**: *Cadophora* (16.09%), *Sporothrix* (12.35%), *Clonostachys* (10.36%), *Penicillium* (10.31%), *Alternaria* (9.45%) *Trichoderma* (8.79%), *Talaromyces* (6.80%), *Diaporthe* (4.47%), *Dictyosporium* (3.94%) *Nothophoma* (3.78%), and *Phialocephala* (3.01%). The most abundant genera **180 days after planting were**
**found**
**in**
**the following**
**untreated plants**: *Meliniomyces* (14.10%), *Clonostachys* (10.75%), *Cadophora* (9.90%), *Alternaria* (8.72%), *Talaromyces* (7.71%), *Trichoderma* (6.31%), *Sporothrix* (5.94%), *Minutisphaera* (5.78%), *Acremonium* (5.07%), and *Kalmusia* (4.42%). The most abundant genera **180 days after planting were**
**found**
**in**
**the following**
**treated plants:**
*Sporothrix* (13.74%), *Talaromyces* (11.52%), *Clonostachys* (10.79%), *Meliniomyces* (9.02%), *Diaporthe* (7.75%), *Nectria* (6.83%), *Phialocephala* (4.76%), *Sarocladium* (4.64%), *Trichoderma* (3.87%), *Penicillium* (3.65%), *Cladosporium* (3.22%), and *Cadophora* (2.38%).

### 3.5. Alpha Diversity Metrics Did Not Differ among Treated × Untreated Plants

The alpha diversity of fungal communities in grapevine wood samples did not differ significantly between treated × untreated plants (Figure 5).

### 3.6. Shared Fungal Assemblages and Unique OTUs

The percentage of shared OTUs among both treatments and both sampling periods was 15.7% (Figure 6, Appendix A). The OTUs that were unique in each treatment and sampling period are shown in (Appendix A). Core taxa with more than 50% prevalence (Appendix A) **10 days after treatment in untreated plants were the following**: *Diaporthe* 100%, *Ectophoma* 100%, *Trichoderma* 83%, *Phaeoacremonium* 83%, *Notophoma* 83%, *Fusarium* 83%, *Cadophora* 83%, *Stenocarpella* 67%, *Alternaria* 67%, *Talaromyces* 50%, *Sporothrix* 50%, and *Penicillium* 50%. Core taxa with more than 50% prevalence **10 days after treatment in treated plants were the following**: *Trichoderma* 100%, *Sporothrix* 100%, *Cadophora* 100%, *Talaromyces* 80%, *Penicillium* 80%, *Clonostachys* 80%, *Alternaria* 80%, *Pseudogymnoascus* 60%, and *Nothophoma* 80%. Core taxa with more than 50% prevalence **180 days after treatment in untreated plants were the following**: *Sporothrix* 100%, *Cadophora* 100%, *Talaromyces* 83%, *Clonostachys* 83%, *Alternaria* 100%, *Trichoderma* 67%, *Phialocephala* 67%, *Penicillium* 50%, *Meliniomyces* 50%, *Fusarium* 50%, and *Cladosporium* 50%. Core taxa with more than 50% prevalence **180 days after treatment in treated plants were the following**: *Cladosporium* 83%, *Trichoderma* 67%, *Talaromyces* 67%, *Sporothrix* 67%, *Penicillium* 67%, *Clonostachys* 67%, *Alternaria* 67%, and *Vishniacozyma* 50%.

### 3.7. The Natural Infection Rates Caused by Fungal Trunk Pathogens Differed between Treated Untreated Plants

Among the identified fungal taxa, seven genera were associated with GTDs: *Cadophora*, *Dactylonectria*, *Diaporthe*, *Diplodia*, *Lasiodiplodia*, *Phaeoacremonium*, and *Phaeomoniella*. The OTU table (Appendix A) reveals very low read counts of the genus *Diplodia* only in two samples within the untreated plants, 180 days after treatment; for the genus *Lasiodiplodia*, we detected very low read counts only in two samples of treated plants 180 days after treatment; the genus *Phaeomoniella* was detected in very low read counts within two samples 180 after treatment and in one sample 10 days after treatment, not in untreated plants. The results for those three genera were not statistically significant, considering the low read counts and low prevalence. The genus *Dactylonectria* was present in very low read counts within both treatments and both sampling periods with no significant differences. The genus *Cadophora* was present within both treatments and sampling periods. Ten days after treatment, there was no significant difference (*p* = 0.4652) between treated × untreated plants. Nonetheless, the LEfSe analyses (Figure 7) revealed significant differences in the genus *Cadophora* (*p* = 0.0250) 180 days after treatment between treated × untreated plants. Treated plants showed lower relative abundances (2.38 vs. 9.90%, Figure 4) and lower prevalence (40 vs. 100%, Appendix A) than untreated plants. The genera *Diaporthe* and *Phaeoacremonium* were significantly discriminated by LEfSe analyses (*Diaporthe p* = 0.0176, *Phaeoacremonium p* = 0.0446) 10 days after treatment. The genus *Diaporthe* showed lower relative abundances in treated plants (4.47 vs. 19.34%) and lower prevalence in samples (40 vs. 100%) compared with untreated plants. A similar result was observed for *Phaeoacremonium*-treated plants, which showed lower relative abundance (0.01 vs. 0.04%) and lower prevalence within samples (20 vs. 83%) than untreated plants. Detailed results of LEfSe analyses discriminating significantly different GTD OTUs between untreated and treated plants are shown in (Appendix A). The genus *Fusarium* was discriminated (*p* = 0.0426) in the variant 10 days after treatment, and treated plants showed a lower relative abundance (0.27%) of this genus than untreated plants (3.65%).

### 3.8. The Non-GTD Mycobiota Varied among Treated × Untreated Plants

Among nonpathogenic species, the LEfSe analyses positively discriminated ten days after treatment from the treated plant genera *Aspergillus* (*p* = 0.0128), *Oidiodendron* (*p*= 0.017365), and *Dictyosporium* (*p* = 0.0111), and negatively discriminated the genus *Ectophoma* (*p* = 0.0176) (Figure 7a). In the second sampling period 180 days after treatment, the LEfSe analyses discriminated positively from the treated plant genus *Nectria* (*p* = 0.0489) and negatively from the genera *Scolecolachnum* (*p* = 0.0222) and *Neocatenulostroma* (*p* = 0.0369) (Figure 7b).

## 4. Discussion

This is the first study unveiling the antifungal effects of a crude knotwood extract from *P. abies* against GTD fungi. Due to the non-effectivity of conventional fungicides/treatments against GTDs [32,33], which have often negative impact on the environment [34,35], there is a need to find alternatives capable of protecting grapevines. Recent studies on knotwood extracts of coniferous trees have been found to have antifungal properties against various fungal species, including *Aspergillus fumigatus*, *Fibroporia vaillantii*, *Gloeophyllum trabeum*, *Penicillium brevicompactum*, *Schizophyllum commune*, and *Trametes versicolour* [36,37,38]. Thus, this study focused on determining the potential antifungal properties of GE against GTDs.

The in vitro experiment revealed high antifungal effects of GE against all tested GTD fungi. Using GE diluted to 1 mg·mL^−1^ HMR in medium, we observed a total inhibition of growth for *P. minimum* and *C. luteo-olivacea*. Approximately 29 species of *Phaeoacremonium* and 6 species of *Cadophora* together with *Phaeomoniella chlamydospora* (W. Gams, Crous, M.J. Wingf. and Mugnai), Crous and W. Gams, and *Pleurostoma richardsiae* (Nannfeldt), Réblová and Jaklitsch, are responsible for Petri disease [8], while *P. minimum* and *C. luteo-olivacea* are the most prevalent causal agents [39,40]. *Phaeoacremonium* spp. can spend a part of their disease cycle in soil/pruning debris and thus represent a potential source of inoculum for noninfected vines [40,41]. The presence of *Cadophora* spp. in soil was confirmed recently by an HTAS study [21], and the presence of both genera was already reported in nurseries producing propagation material in Algeria, Spain, the Czech Republic, South Africa, Canada, and the USA [21,42,43,44,45,46,47]. Total inhibition of one of the major causal agents of black foot disease—*Da. torresensis* was also observed. Black foot disease is a soil-borne disease caused by *Cylindrocarpon*-like fungi [9]. Among these, *Da. torresensis* is the most frequently isolated species [48,49,50,51,52,53], already reported from young vineyards in Spain, Australia, Canada, the Czech Republic, New Zealand, Portugal, South Africa, Spain, and Turkey [49,51,52,53], and from symptomless grapevine nursery stock in Spain [54]. The growth of the other four tested fungi (*D. bohemiae*, *D. ampelina*, *Dip. seriata*, *E. lata*) was not completely inhibited using GE diluted to contain 1 mg·mL^−1^ HMR in medium. However, partial inhibition of growth was observed against all four pathogens. *Diaporthe bohemiae* (Guarnaccia, Eichmeier, and Crous) was described for the first time in 2018 from the Czech Republic [10], and its pathogenicity on grapevine was confirmed recently [55]. An in vitro trial revealed 58.4% inhibition of this pathogen using GE. The second tested *Diaporthe* species, *D. ampelina*, was inhibited by 62.5% in vitro. *Diaporthe ampelina* represents a causal agent of Phomopsis cane and leaf spot reported from vineyards worldwide [56,57,58]. *Eutypa lata*, a worldwide causal spread agent of Eutypa dieback [59,60,61,62,63,64], was inhibited by 79% using GE in vitro. *Diplodia seriata*, member of the family *Botryosphaeriaceae* and the worldwide causal spread agent of Botryosphaeria dieback on grapevine [65,66,67,68,69], was the least inhibited fungus using GE in vitro, with inhibition of growth equal to 30%. Recent studies reported antifungal activity of pine bark extracts against fungal pathogens in *Botryosphaeriaceae* [70,71].

In general, the intensity of inhibition was associated with the concentration of GE in agar media the rate of inhibition increased with increasing concentration. Among the three tested concentrations of 0.1, 0.5, and 1 mg·mL^−1^, the concentration of 1 mg·mL^−1^ was the most effective. In most experiments, HMR^TM^ lignan alone provided lower in vitro antifungal activity than GE. The higher antifungal effects of GE against most GTDs compared to pure HMR^TM^ observed in our study are consistent with a study comparing the antifungal effects of phenolic extracts from *P. abies* and its main compounds (HMR, Coni) against the fungus *Heterobasidion annosum* [16]. The only exception was observed against the genus *Diaporthe*. In the case of *D. ampelina*, the HMR ^TM^ plates provided stronger inhibition (5%) than the GE plates. For *D. bohemiae*, the results were similar for both the GE and HMR^TM^ treatments, with no statistical significance. The stronger antifungal activity of GE against most of the tested fungal species compared to pure HMR^TM^ is probably caused by other compounds that are present in minor volumes within GE, e.g., other lignans, oligolignans, stilbenes, terpenes, or flavonoids [16,20,72,73,74,75], that could improve the antimicrobial effect of HMR^TM^ lignan.

Based on promising in vitro results, we decided to evaluate the antifungal effects of GE on grapevine plants. Grapevine is one of the frontier plants studied by omic technologies, and its fungal communities have been studied extensively [76,77,78,79,80,81,82,83,84,85,86]. Recent studies used HTAS as a detection tool in plant protection experiments, e.g., how grapevine pathogens/mycobiota are affected by pruning time [87], hot water treatment [21], antagonistic bacteria [88], biocontrol agents [89], or different fungicides [90]. In our experiment, HTAS was used to explore the antifungal effects of GE treatment against fungi colonizing young grapevine plants in two periods after the treatment. The HTAS results showed that the GE treatment did not significantly affect the fungal microbial alpha and beta diversities of treated plants compared to untreated plants, probably because wood extractives do not control the growth of a broad spectrum of fungal species but may be quite specific [91]. Ascomycota was the predominant fungal phylum for both treatments and sampling periods, followed by Basidiomycota. The fungal phyla compositions were consistent with recent HTAS studies on grapevines [21,22,77,84,87]. The core mycobiome descriptions of both treated/untreated plants were similar and consisted of the fungal genera *Alternaria*, *Cadophora*, *Clonostachys*, *Diaporthe*, *Penicillium*, *Sporothrix*, *Trichoderma*, and *Talaromyces*. Among the saprophytes and endophytes, the genera *Alternaria*, *Cladosporium*, and *Penicillium* were already reported from a study characterizing the mycobiome of rootstocks at different stages of the grapevine propagation process [92], and the genera *Clonostachys*, *Talaromyces*, and *Trichoderma* were already reported in a study examining the biocontrol potential of grapevine endophytes in Portugal [93]. Recently, the antifungal activity of wood extracts from *P. abies* was confirmed against *Aspergillus* spp. (*A. flavus* Link, *A. ochraceus* K. Wihl, *A. niger* Tiegh.) in vitro [13]. That is, in contrast with our HTAS result, the genus *Aspergillus* was more abundant in plants treated by GE ten days after treatment. Among GTDs, the HTAS results supported the in vitro inhibitory effects of GE against the genera *Cadophora*, *Diaporthe*, and *Phaeoacremonium*. In the case of *C. luteo-olivacea*, we observed complete inhibition of growth in vitro that corresponds to significantly lower abundances (76% lower) of the genus *Cadophora* in grapevines treated with GE compared with untreated plants 180 days after the treatment. A similar scenario was observed for the genera *Diaporthe* and *Phaeoacremonium;* treated plants showed lower relative abundances (76.6% and 80% lower) of both pathogenic genera than untreated grapevines 10 days after treatment. *Fusarium* spp. have already been described as root pathogens causing a decline of vines in reports from Australia, Egypt, Japan, and Sicily [94,95,96,97,98]; on the other hand, several studies have proposed *Fusarium* spp. as a potential biocontrol agent against GTDs [93,99,100]. In our study, we observed a 92.6% lower relative abundance of *Fusarium* in treated plants than in untreated plants ten days after treatment.

The results of our study confirm the antifungal effects of GE and the presence of GTD pathogens in Czech nurseries, which were reported in 2018 [21]. These results are consistent with those of recent studies examining GTD pathogens in other nurseries around the world, including Australia, France, New Zealand, Portugal, South Africa, Spain, and Uruguay [101,102,103,104,105,106]. The mentioned studies describe the spread of GTDs in nurseries worldwide and declare the urgent need to find a solution to the current situation. Producing vital and healthy propagation material is an essential step to prevent GTD development [107]. The wood extracts from *P. abies* were already proposed to be a possible agricultural pest control tool in 2016 [13]. In addition to current preventive techniques based on various chemicals, biocontrol agents, and pruning management [108,109,110,111,112,113,114,115,116,117,118], GE could help to produce healthy vines. However, additional tests still have to be performed.

## 5. Conclusions

This study provides guiding insight into a bioprotection strategy using a knotwood extract from spruce trees against GTDs. Presented study confirmed the in vitro antifungal effects of GE against representative pathogens of the GTD complex. Complete inhibition of growth was observed against *C. luteo-olivacea*, *Da. torresensis*, and *P. minimum*. Partial inhibition was observed against the fungi *D. ampelina* (62.5%), *D. bohemiae* (58.4%)*, Dip. Seriata* (30%), and *E. lata* (79%) using 1 mg·mL^−1^ extract. The efficiency of inhibition was related to the concentration of GE in the agar plates. The rate of inhibition increased with increasing concentrations of GE. Among the three tested concentrations (0.1, 0.5, 1 mg·mL^−1^), the highest tested concentration of 1 mg·mL^−1^ was the most effective against all fungi.

Among the HTAS results, GE-treated plants showed lower relative abundances of genera *Diaporthe* and *Phaeoacremonium* than untreated plants ten days after treatment. This trend was not confirmed in *Diaporthe* 180 days after treatment. Currently, we are not able to say whether GE was effective in the long-term trial against *Phaeoacremonium* or if the genus disappeared naturally because this genus was not detected in either treated or untreated plants 180 days after planting. The most promising result in planta was observed in the long-term trial in genus *Cadophora.* Treated plants showed significantly lower abundances and prevalence of the genus *Cadophora* than untreated plants 180 days after planting.

For future studies, we recommend evaluating inhibition effects against additional fungal species in vitro considering plant benefit species, e.g., *Trichoderma*. We also recommend using a concentration of GE higher than the 1 mg·mL^−1^ HMR used in these trials, which could possibly increase the inhibition effects. The present study used naturally infected plants, among which the incidence of GTD fungi varied at the start of the experiment. In future experiments, it will be necessary to evaluate the inhibitory effects of GE in planta using an inoculation technique combining GE treatment and particular GTD fungal strains.

## Figures and Tables

**Figure 1 jof-08-00357-f001:**
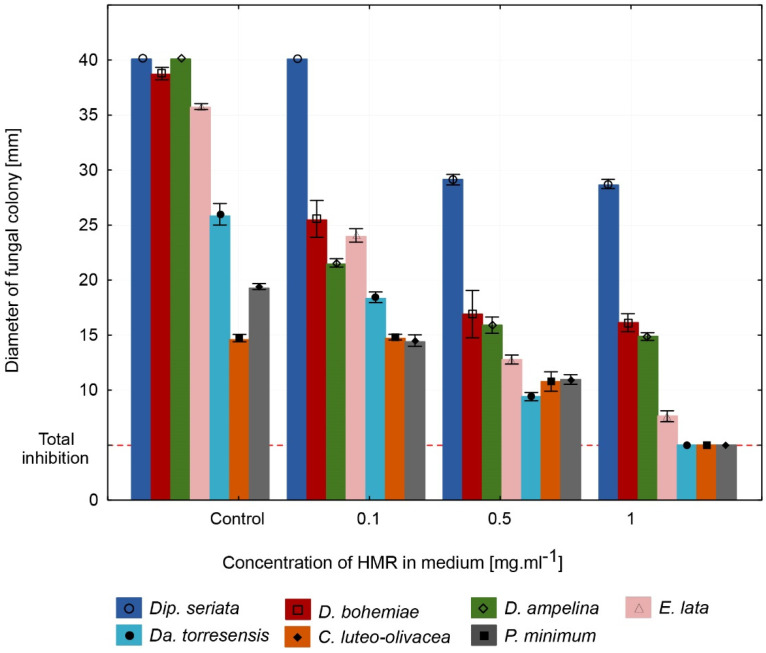
Antifungal effect of GE after 5 days of cultivation for all tested fungi except *P. minimum* and *C. luteo-olivacea*, which were measured after 10 days. A colony diam. of 5 mm was considered to be a total inhibition of growth, since it was the size of the inoculation disk.

**Figure 2 jof-08-00357-f002:**
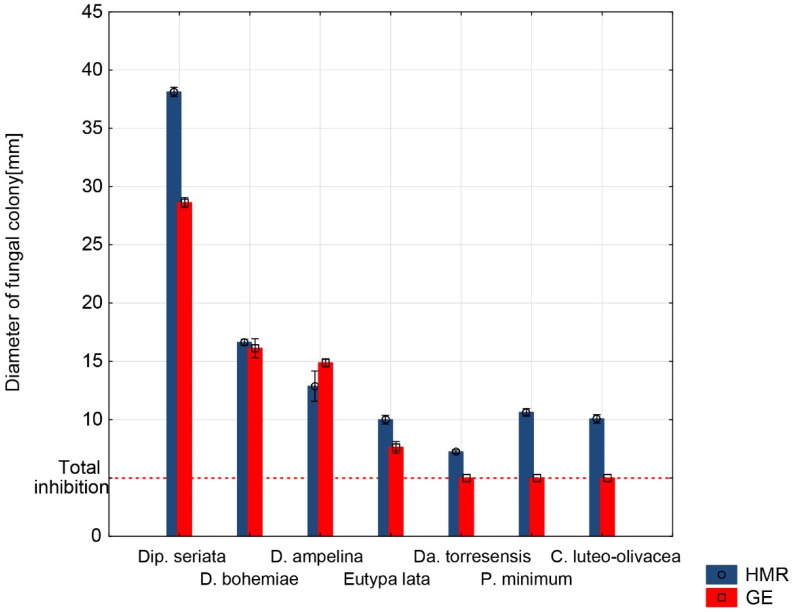
Comparison of antifungal effects of pure HMR^TM^ and GE; plates of both variants contained 1 mg·mL^−1^ HMR in the medium. Measurements were performed after 5 days of incubation for all fungi except *P. minimum* and *C. luteo-olivacea*, which were measured after 10 days. A colony diam. of 5 mm was considered to be a total inhibition of growth since it was the size of the inoculation disk.

**Figure 3 jof-08-00357-f003:**
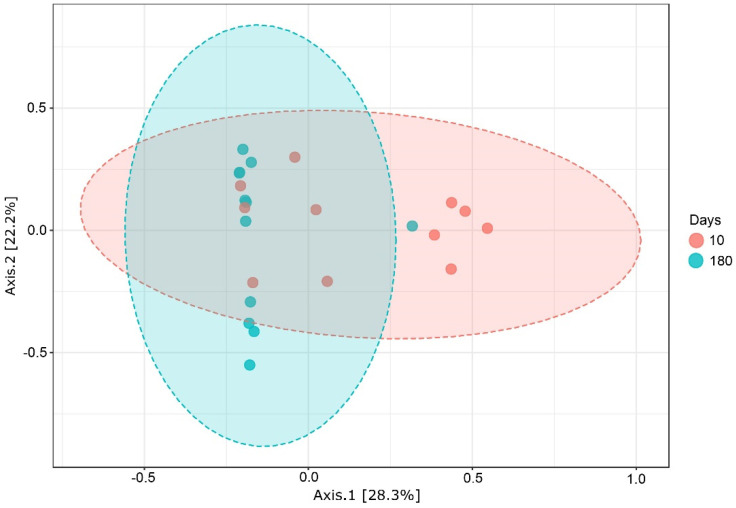
Principal coordinate analysis (PCoA) results based on Bray-Curtis dissimilarity metrics in 2D, showing the distance in the fungal communities among the 2 terms of sampling—10 days and 180 days after treatment.

**Figure 4 jof-08-00357-f004:**
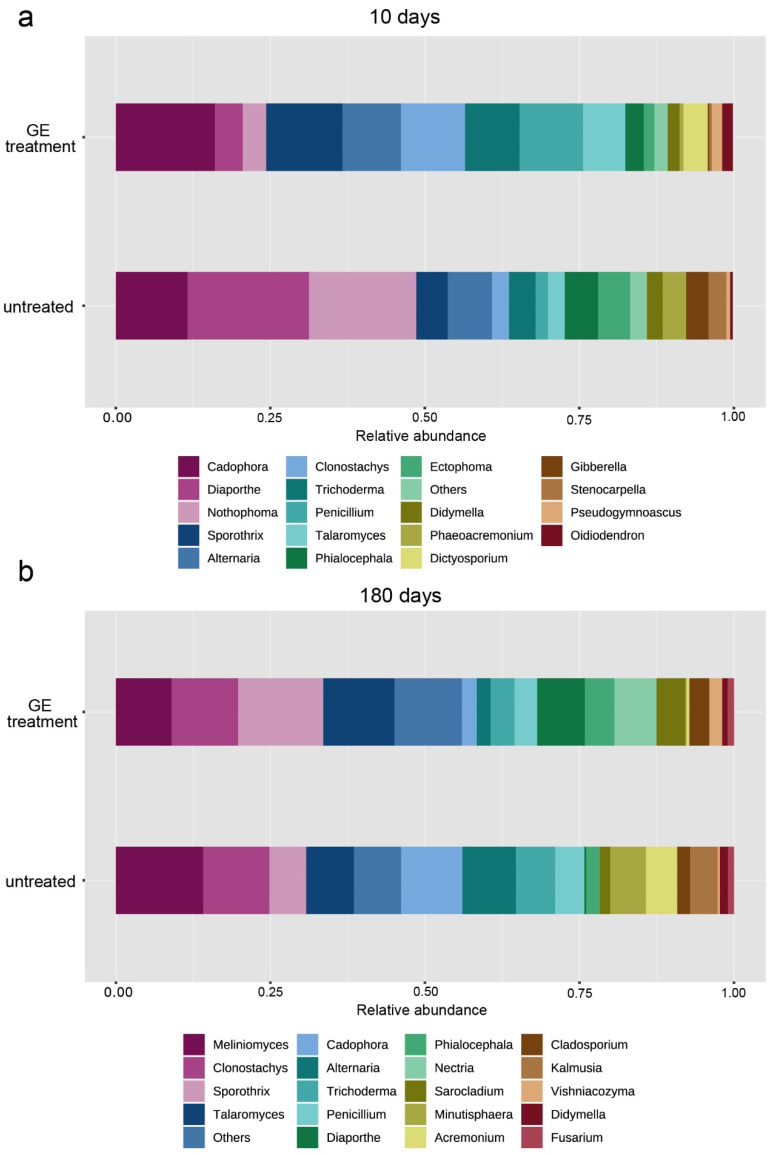
Relative abundances of the different fungal genera in plants treated by GE × untreated plants in two terms after sampling: (**a**) 10 days after treatment; (**b**) 180 days after treatment.

**Figure 5 jof-08-00357-f005:**
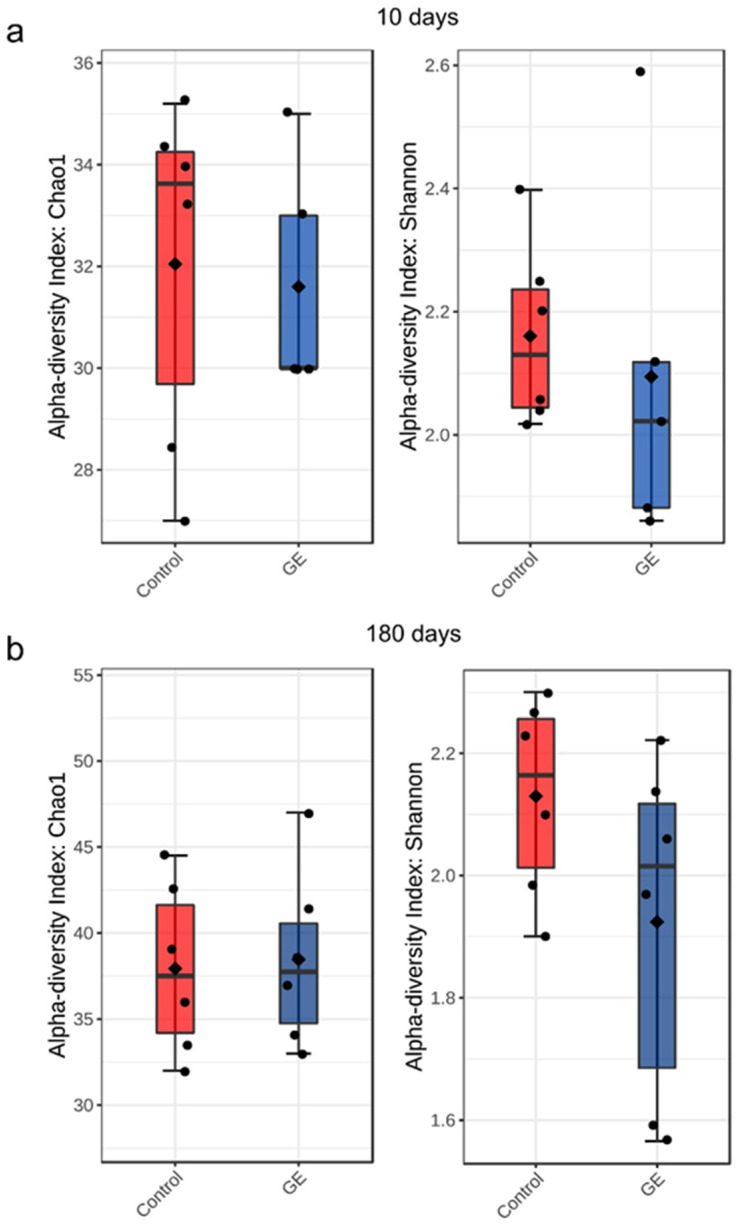
Boxplots illustrating differences in Chao1 and Shannon diversity measures of the fungal communities between treated × untreated plants: (**a**) 10 days after treatment; (**b**) 180 days after treatment; GE—treated plants; Control—untreated plants.

**Figure 6 jof-08-00357-f006:**
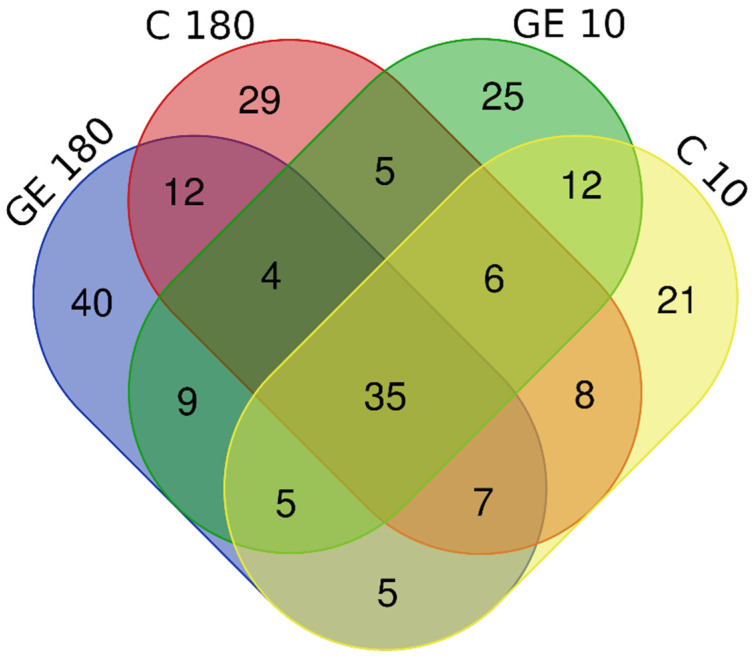
Venn diagram displaying shared OTUs among treatments and sampling periods: **GE 180**—treated plants 180 days after treatment; **C 180**—untreated plants 180 days after treatment; **GE 10**—treated plants 10 days after treatment; **C 10**—untreated plants 10 days after treatment.

**Figure 7 jof-08-00357-f007:**
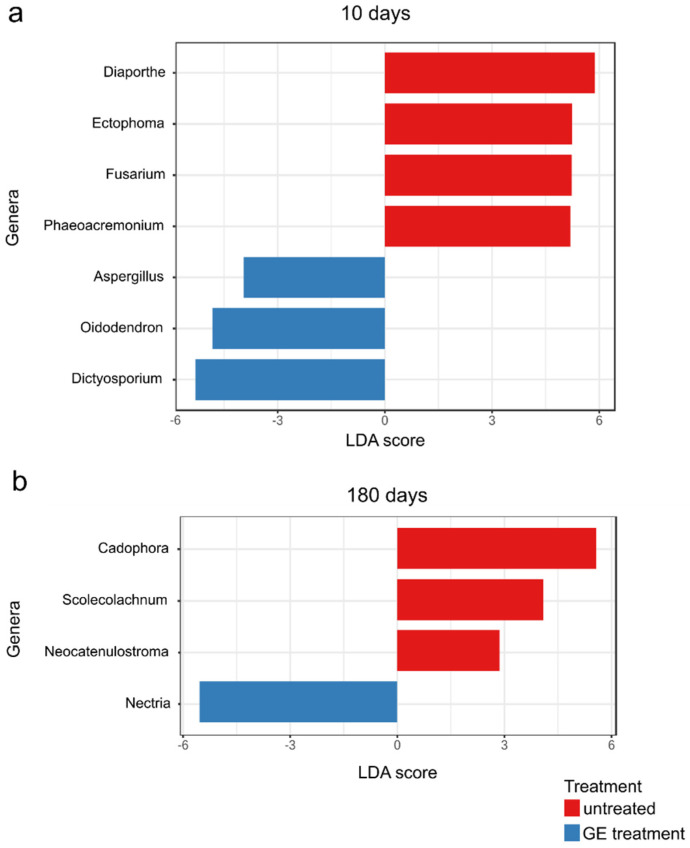
LEfSe analyses discriminating significantly different OTUs between treated (GE)/untreated plants: (**a**) 10 days after treatment; (**b**) 180 days after treatment.

## Data Availability

Not applicable.

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
