# Peer review of "Lignans Extract from Knotwood of Norway Spruce—A Possible New Weapon against GTDs"

_jof, 2022, doi:10.3390/jof8040357_

Round 1
Reviewer 1 Report
I revised the manuscript by Spetik et al. I think it is a nice piece of work though I thought some parts could be improved for clarity and others need some further explanation. The most controversial (and interesting) part of the paper is when testing the effects of the treatments in the field. The authors claim that there are no significant differences for Cadophora (ca. p=0.49) but then they use a test that I myself do not understand and the results are significant. I think the authors should critically revise how that test LEfSe works and how come it shows such significant results. One possibility is that the data is not normal and therefore using normal ANOVA testing one cannot see differences when there are. In that case I would recommend them to use some sort of transformation of using generalised model. I am saying all that because when one reads the abstract it looks as there are some really strong effects of the GE treatment, but then when reading the paper one sees that the effects are a bit depending on the test and that perhaps results are not that sharp.
Other aspects to consider refer to the Figures. In most of them there is a mix of fonts, Times, calibri, arial, etc. I would suggest using one type preferably arial all over. The figures are in my opinion a bit disproportioned (often too large). The numbers in the axis are very small, sometimes the legend is hard to read. There is a slight abuse of colours.
Figure 2 can be reduced in height, and font size can be doubles at least. The same for figure 1, although these two figures are the best ones.
Figure 3 has very small axis titles, units and the legend is very small too. It would be nice if all graphs have somehow the same style, and even they use the same colour if blue and red is used for treatment and control, the same colours could be used all over the paper.
Figure 4 is very hard to read as there are many colours in the same bar. The legend is confusing and the names are too small. It is hard to compare the size of the smallest bars, or of the bars that are not perfectly aligned to each other. It is also not super clear what a "detected in treated (GE) × untreated plants " means as it is spelled in the legend.
Figure 6 with the venn diagram is very difficult to read. There are so many combinations and actually I am not sure that the information one gets from there are so important. At the end, what really matters from this research is if the treatment works or not, and on which fungi (if it would not work it would be a very good study too). So to me all what is mentioned in 3.6% is rather hard to follow and could be deleted.
Figure 5 is really huge. I think it can be reduced a lot and show the same. GE should be explained in the legend. Using the same colours for control and treatment would make it easier to understand.
Figure 7 is strange because both boxes are the same size but the upper one has many more bars. If the same bar size would be used, one would realise that there are less fungi showing differences at 180 days than at 10 days.
In the the part 3.8 the p values are very long, i.e. "(P=0.012799)" it would be better if the same number of decimals is consistently used across the text.
In l. 445 the authors speak about biocontrol. I am not sure about what definition of biocontrol they refer but to my understanding biocontrol would only imply the use of living organisms to control diseases:
Stenberg, J.A., Sundh, I., Becher, P.G. et al. When is it biological control? A framework of definitions, mechanisms, and classifications. J Pest Sci 94, 665–676 (2021). https://doi.org/10.1007/s10340-021-01354-7
Across the manuscript there are many instances where it shows: (Error! Reference source not found.).
Both Discussion and Conclusion are written as big captions. I would suggest dividing the text in smaller captions to improve readibility.
In line 443, the authors claim that further tests should be performed. Perhaps they could be more specific and mention which ones would be those tests?
in line 342 and 343 there is a wierd line jump
I would like to thank the authors for such an interesting and complete experiment. I hope the comments help them improve and give the final facelift.
Author Response
Questions
Response
The most controversial (and interesting) part of the paper is when testing the effects of the treatments in the field. The authors claim that there are no significant differences for Cadophora (ca. p=0.49) but then they use a test that I myself do not understand and the results are significant. I think the authors should critically revise how that test LEfSe works and how come it shows such significant results. One possibility is that the data is not normal and therefore using normal ANOVA testing one cannot see differences when there are. In that case I would recommend them to use some sort of transformation of using generalised model. I am saying all that because when one reads the abstract it looks as there are some really strong effects of the GE treatment, but then when reading the paper one sees that the effects are a bit depending on the test and that perhaps results are not that sharp.
For the LEfSe analyses we split the dataset based on term of sampling because the sampling term was the biggest source of diversity between samples. The first dataset contained samples10 days after treatment and the second set contained data from samples 180 days after treatment. This is the reason why there are two P values. Based on LEfSe analyses the results were significant only for second term (180 DaT) P=0.02497 and not for the first one P=0.4652 (10 DaT).
We interpreted these results in the way that the extract may be not effective immediately but needs some time to inhibit the growth of Cadophora.
Other aspects to consider refer to the Figures. In most of them there is a mix of fonts, Times, calibri, arial, etc. I would suggest using one type preferably arial all over. The figures are in my opinion a bit disproportioned (often too large). The numbers in the axis are very small, sometimes the legend is hard to read. There is a slight abuse of colours.
Figure 2 can be reduced in height, and font size can be doubles at least. The same for figure 1, although these two figures are the best ones.
Thank you very much for this comment, we corrected it according to the comments.
Figure 3 has very small axis titles, units and the legend is very small too. It would be nice if all graphs have somehow the same style, and even they use the same colour if blue and red is used for treatment and control, the same colours could be used all over the paper.
Reworked according to the comment.
Figure 4 is very hard to read as there are many colours in the same bar. The legend is confusing and the names are too small. It is hard to compare the size of the smallest bars, or of the bars that are not perfectly aligned to each other. It is also not super clear what a "detected in treated (GE) × untreated plants " means as it is spelled in the legend.
Reworked according to the comment.
Figure 6 with the venn diagram is very difficult to read. There are so many combinations and actually I am not sure that the information one gets from there are so important. At the end, what really matters from this research is if the treatment works or not, and on which fungi (if it would not work it would be a very good study too). So to me all what is mentioned in 3.6% is rather hard to follow and could be deleted.
Venn diagram is usually used as visualisation of metagenomic studies like our study. We prefer to keep the figure.
Figure 5 is really huge. I think it can be reduced a lot and show the same. GE should be explained in the legend. Using the same colours for control and treatment would make it easier to understand.
Reworked according to the comment.
Figure 7 is strange because both boxes are the same size but the upper one has many more bars. If the same bar size would be used, one would realise that there are less fungi showing differences at 180 days than at 10 days.
Reworked according to the comment.
In the part 3.8 the p values are very long, i.e. "(P=0.012799)" it would be better if the same number of decimals is consistently used across the text.
Corrected according to the comment.
In l. 445 the authors speak about biocontrol. I am not sure about what definition of biocontrol they refer but to my understanding biocontrol would only imply the use of living organisms to control diseases:
Stenberg, J.A., Sundh, I., Becher, P.G. et al. When is it biological control? A framework of definitions, mechanisms, and classifications. J Pest Sci 94, 665–676 (2021). https://doi.org/10.1007/s10340-021-01354-7
We replaced “biocontrol” with “bioprotection”, thank you very much for this comment.
Across the manuscript there are many instances where it shows: (Error! Reference source not found.).
During the submission the supplementary files were included within the manuscript. The supplementary data were probably removed from the main document before for the review process started. It led to errors in hyperlinks referring to supplementary files. Based on reviewers comments the hyperlinks were replaced by text (Figure S2, Figure S3., etc.).
Both Discussion and Conclusion are written as big captions. I would suggest dividing the text in smaller captions to improve readibility.
Corrected according to the comment.
In line 443, the authors claim that further tests should be performed. Perhaps they could be more specific and mention which ones would be those tests?
It is mentioned in the chapter conclusion: “In future experiments, it will be necessary to evaluate the inhibitory effects of GE in planta using an inoculation technique combining GE treatment and particular GTD fungal strains.” We suppose that it is enough in that moment.
in line 342 and 343 there is a wierd line jump
This is probably due to a PDF conversion in the .docx file the problem does not exist.
I would like to thank the authors for such an interesting and complete experiment. I hope the comments help them improve and give the final facelift.
We thank to reviewer a lot for these helpful comments.

Reviewer 2 Report
- Objective 3 in line no. 59-60 is not clear. Is it for control ?
- Line no. 219, 231, 234, 235 and other places: (Error! Reference source not found.). -What does it mean.
Author Response
Questions
Response
1.Objective 3 in line no. 59-60 is not clear. Is it for control?
It is one point of our general hypothesis that is not associated with the knotwood extract. This hypothesis was already confirmed in our previous results published by Eichmeier et .al 2018.
Eichmeier, A.; Pečenka, J.; Peňázová, E.; Baránek, M.; Català-García, S.; León, M.; Armengol, J.; Gramaje, D. High-throughput amplicon sequencing-based analysis of active fungal communities inhabiting grapevine after hot-water treatments reveals unexpectedly high fungal diversity. Fungal Ecology 2018, 36, 26-38, doi:10.1016/j.funeco.2018.07.011.
2.Line no. 219, 231, 234, 235 and other places: (Error! Reference source not found.). -What does it mean.
During the submission the supplementary files were included within the manuscript. The supplementary data were probably removed from the main document before for the review process started. It led to errors in hyperlinks referring to supplementary files. Based on reviewers comments the hyperlinks were replaced by text (Figure S2, Figure S3., etc.).
Reviewer 3 Report
The revision has been carried out using track mode in the attached pdf file .

Author Response
We would like to thank reviewer for the comments. All questions were answered within the PDF file.

Reviewer 4 Report
The manuscript "Lignans Extract from Knotwood of Norway Spruce—A New
3 Weapon against GTDs?" has been reviewed for publication in JoF. The work represents an interesting contribution to the search for new natural alternatives for the control of so-called GTDs in grapevine crops. Lignan extract is one of those compounds with potential for its application as a substitute for chemically synthesized fungicides. The work is well structured and the methodology used at the different test levels is correct for the intended objectives. However, one of the main weaknesses of the work (assumed some way by the authors themselves) is the fact that they have worked (at the level of field trials) with plant material not previously analyzed, of which the degree of infection and specific composition of the different etiological agents of GTDs present in the grafted plants from the nursery before their treatment are unknown. Therefore, to draw robust conclusions about the ability to reduce the presence of GTDs in the field from the treatments carried out seems risky.
Some additional observations / suggestions have been made directly in the attached pdf

Author Response
We would like to thank reviewer for helpful comments and revisions. All questions were answered directly within the PDF file. The corrections of spelling and formatting were all accepted and implemented within the .docx manuscript.
